# A Novel Evaluation Method for SLAM-Based 3D Reconstruction of Lumen Panoramas

**DOI:** 10.3390/s23167188

**Published:** 2023-08-15

**Authors:** Xiaoyu Yu, Jianbo Zhao, Haibin Wu, Aili Wang

**Affiliations:** 1College of Electron and Information, University of Electronic Science and Technology of China, Zhongshan Institute, Zhongshan 528402, China; yuxy@zsc.edu.cn; 2Heilongjiang Province Key Laboratory of Laser Spectroscopy Technology and Application, Harbin University of Science and Technology, Harbin 150080, Chinaaili925@hrbust.edu.cn (A.W.)

**Keywords:** monocular vision, simultaneous localization and mapping, 3D reconstruction, accuracy evaluation

## Abstract

Laparoscopy is employed in conventional minimally invasive surgery to inspect internal cavities by viewing two-dimensional images on a monitor. This method has a limited field of view and provides insufficient information for surgeons, increasing surgical complexity. Utilizing simultaneous localization and mapping (SLAM) technology to reconstruct laparoscopic scenes can offer more comprehensive and intuitive visual feedback. Moreover, the precision of the reconstructed models is a crucial factor for further applications of surgical assistance systems. However, challenges such as data scarcity and scale uncertainty hinder effective assessment of the accuracy of endoscopic monocular SLAM reconstructions. Therefore, this paper proposes a technique that incorporates existing knowledge from calibration objects to supplement metric information and resolve scale ambiguity issues, and it quantifies the endoscopic reconstruction accuracy based on local alignment metrics. The experimental results demonstrate that the reconstructed models restore realistic scales and enable error analysis for laparoscopic SLAM reconstruction systems. This suggests that for the evaluation of monocular SLAM three-dimensional (3D) reconstruction accuracy in minimally invasive surgery scenarios, our proposed scheme for recovering scale factors is viable, and our evaluation outcomes can serve as criteria for measuring reconstruction precision.

## 1. Introduction

A precise three-dimensional reconstruction model that incorporates internal anatomical structures and laparoscopic positions can assist surgeons in making decisions and performing operations. Furthermore, the reconstruction of a surgical scene is an essential step for data registration in surgical navigation and surgery with augmented reality. As an approach to this, vision-based simultaneous localization and mapping (VSLAM) in miniature medical devices has gained interest [1]. By applying three-dimensional SLAM reconstruction techniques, one can obtain the three-dimensional structures and shapes of relevant targets, which can provide prior information for spatial target navigation without configuration information, can be used to capture the structural dimensions of target bodies, environments, and regions of interest, and can be used to estimate the motion states of navigation robots. Using SLAM techniques to achieve three-dimensional endoscopic reconstruction from sequential images acquired via laparoscopy is a convenient and effective way to obtain environmental information and locate endoscopes [2,3,4].

The accuracy of three-dimensional reconstruction has a direct impact on the outcomes of subsequent research and applications. Evaluating the accuracy of three-dimensional reconstruction is not only for verifying the effectiveness and robustness of reconstruction algorithms and assessing the precision and completeness of reconstruction results, but is also more importantly for providing crucial references for further three-dimensional analysis and visualization. However, due to the specificity of application scenarios, it is challenging to acquire ground-truth data for three-dimensional endoscopy in minimally invasive surgery. Because different parts of endoscopy have varying degrees of complexity, as well as distinct features among the various organ tissues involved, it is hard to establish comprehensive three-dimensional endoscopic testing data; therefore, the main method still depends on other medical devices for measuring relevant morphological data [5]. However, it is not easy to obtain real 3D data on a target location in an internal cavity in real medical scenarios. If real data are collected from an internal cavity by introducing small 3D scanning devices, firstly, it is necessary not only to meet the surgical requirements and safety requirements of minimally invasive surgery, but also to consider the patient’s physical tolerance when collecting data. Secondly, this also involves corresponding privacy and ethical issues. Furthermore, the size, difficulty, and cost of such devices are also problems that cannot be overlooked. Under special circumstances, such as when patients’ objective conditions are poor and require a certain degree of reduction in the harm caused by ionizing radiation, external collection methods cannot meet the requirements of conventional inspection with a CT or MRI device, therefore leading to an inability to use such devices for the collection of 3D data.

Moreover, the main disadvantage of monocular SLAM is that the absolute scale of the reconstructed scene and the camera movement are intrinsically unobservable. Without additional metric information, the 3D reconstruction results of a SLAM system are unclear. Typically, monocular SLAM methods estimate the absolute scale by acquiring extra metric information from distance measurements or an inertial measurement unit (IMU) [6]. Methods that use IMU measurements are termed visual–inertial fusion methods because they combine visual and inertial measurements to estimate camera motion and scene structure [7,8,9]. An accelerometer provides metric information about a robot’s motion. In the methods that use distance measurements, the usual approach is to use depth sensors to measure the distance to a robot’s surrounding environment [10], such as RGB–depth map (RGB-D) [11] cameras or laser scanners [12]. Although these methods have been shown to produce good outcomes, a common shortcoming is that they need extra sensors and hardware to be incorporated into a front-end acquisition system. These extra sensors have their own flaws and restrictions for application in small medical devices, such as space limitations, cost factors, an incompatible design, and biocompatibility issues. They usually require a suitable initial state and precise multi-sensor calibration to prevent convergence to suboptimal local minima. Recently, various deep learning methods [13,14,15] for restoring the global scale of scenes have emerged, but they mostly depend on having real data labels for training. However, acquiring endoscopic datasets and their associated ground-truth depths is challenging. Furthermore, insufficient lighting in a cavity, dynamic and complicated environments, and their reliance on inter-frame photometric information render self-supervised methods rather unstable in their performance.

The quality evaluation of the final modeling output of SLAM-based 3D reconstruction is typically divided into two categories: qualitative and quantitative evaluation. Qualitative evaluation involves a visual comparison, where human perception is used as a subjective indicator to assess the overall visual effect, as well as the details, edges, and other structural information within the reconstructed data. Quantitative evaluation, on the other hand, typically involves an objective assessment of the accuracy of the 3D reconstruction model based on errors and similarities in the data structure. To provide technical support for subsequent engineering applications and further research, quantitative evaluation of the accuracy of 3D reconstructions is particularly important in order to accurately reflect a system’s precision through corresponding error calculations. In special circumstances, such as when a patient’s objective physical conditions are poor and it is necessary to reduce the harm caused by ionizing radiation to a certain extent, external collection methods cannot meet the examination requirements of conventional CT or MRI equipment, resulting in the inability to use such equipment to collect three-dimensional data.

In terms of quantitative evaluation for SLAM-based 3D reconstruction, it has been shown that high accuracy in camera trajectory estimation does not necessarily mean high-quality surface reconstruction. This means that trajectory errors cannot directly express the mapping accuracy of the system. Currently, the evaluation of 3D reconstruction results typically involves statistical analysis based on measured and reconstructed data, which means calculating the error between the reconstructed model and the ground truth model, followed by corresponding analysis and evaluation.

After obtaining the three-dimensional ground truth of the target object, it can be used as a benchmark for quantitative evaluation of the accuracy of reconstructed 3D model. In terms of evaluating the accuracy of the point cloud model obtained from SLAM-based three-dimensional reconstruction, two main indicators are selected for statistical measurement.

The overall absolute accuracy evaluation of the reconstructed point cloud model relies on obtaining high-precision point cloud ground truth using three-dimensional measurement instruments as a reference model. The point cloud obtained from SLAM-based three-dimensional reconstruction is transformed to the same reference coordinate system as the ground truth using manual or ICP algorithms. After alignment with the reference model, the error between the two models is calculated based on Euclidean distance, and the evaluation process flowchart is shown in Figure 1. The distance between the two model is defined as the distance between each point in the reconstructed point cloud and its nearest neighbor in the reference model. The closer the reconstructed point is to the ground truth point, the smaller the distance error and the higher the reconstruction accuracy. The root mean square error (RMSE) is selected to evaluate the overall error of the three-dimensional reconstruction, which represents the oscillation degree of the error and reflects the degree of deviation between the three-dimensional reconstructed point cloud and the ground truth, and characterizes the overall accuracy level of the reconstructed point cloud model from the SLAM-based system.

This paper investigates and examines the prevalent methods for assessing the accuracy of SLAM three-dimensional reconstruction, and addresses the issue of evaluating the precision of cavity-reconstruction point cloud in real-world applications. Without adding extra equipment or sensors, this paper proposes two methods based on the ground control point (GCP) concept to employ scene calibration objects as prior knowledge to augment metric information and restore monocular SLAM scale factor, and quantify them with local alignment degree. The experimental results demonstrated that the reconstructed model restored the true scale information, revealed the overall accuracy of a laparoscopic SLAM system, and offered technical guidance for subsequent applications.

## 2. Evaluation of Accuracy of Inner-Cavity SLAM Reconstruction

To address the evaluation problem of 3D reconstruction in inner-cavity visual SLAM, using real data of human inner cavities has limitations and is difficult to fully test during the research stage. Therefore, based on the actual needs of inner-cavity scenes and limitations such as the equipment and data collection, this paper constructed a simulated inner-cavity experimental platform and used a simulation model to test the 3D reconstruction of inner cavities in a real indoor scene using an inner-cavity monocular SLAM system. In order to recover the scale factor of the inner-cavity monocular SLAM system and conduct accuracy evaluation, this paper combined two feasible methods: intra-cavity calibration object co-reconstruction and external pre-calibration, and mainly calculated the measurement error of the inner-cavity SLAM from the perspective of the absolute size error, and used the root-mean-square error (RMSE) to statistically calculate this error value. During surgery, inspection through laparoscopy focuses more on local measurement information of target areas or specific lesions. After obtaining the reconstructed point cloud model, typical areas of the target organ or soft tissue surface that are significant in the inner-cavity are selected as the target lesion area. Local conformity evaluation is adopted, and the quantitative evaluation of the system’s 3D reconstruction accuracy is carried out from the direction of size conformity of the feature structure.

### 2.1. Intracavitary Calibrator Reconstruction Method

Before calculating the error between the reconstructed measurements and the real size, the idea of the GCP (ground control point) based on the significant ground control points is adopted, and the coordinate transformation relationship is corrected by establishing the coordinate transformation relationship in the base image data with precise information, so that the a priori knowledge obtained by other methods (such as conventional measurements) can be used to establish the estimation relationship, and the point cloud model obtained from the reconstruction coordinates to the world coordinates with real scale. In the 3D reconstruction of the simulated internal cavity, the conversion relationship between the original coordinate system of the reconstructed model and the real-scale world coordinate system is obtained by the known conversion relationship between the actual dimensions of the calibrated object and its dimensions in the 3D point cloud model, so that the 3D reconstruction results can be corrected for the relative scale. Therefore, for the scale uncertainty problem between the reconstructed estimated size of the target object size in the environment and the actual true size, the scale scaling factor *s* is defined as follows:(1)$$s=\frac{d_{i}^{{}^{\prime }}}{d_{i}}$$
where $d_{i}^{\prime }$ and $d_{i}$ are the unitless quantized values of the specific object dimensions in the reconstructed model and the unitary quantized values of their corresponding object dimensions in the actual scene, respectively.

The intracavity calibrator reconstruction method uses the calibration template in the scene to align the 3D reconstruction results to the reconstructed coordinates with standard dimensions, i.e., the 3D coordinates of specific feature points in the reconstruction data under the real space are obtained, and then the dimensions of the target lesion in the internal cavity can be calculated, and the real structure dimensions measured in its reality are calculated and counted, and the dimensional fit of the reconstruction results of the internal cavity target can be obtained.

In the 3D point cloud data-measurement analysis, PCL is used to extract the feature corner points of the target organ tissue surface in the map model using a manual point selection method for measurement. For example, the longest diameter of an irregular lesion can be obtained by using the lesion contour endpoints through the Euclidean distance calculation.

The evaluation index RMSE is calculated as shown in the following Equation (2), where $D_{i}\;$is the measurement value of the corresponding target in the 3D model obtained by fitting the reconstructed point cloud, $D_{i}^{gt}$ is the real size of the actual model, and *N* is the number of measurements.
(2)$$RMSE=\sqrt{\frac{1}{N}\sum_{i=1}^{N}\|D_{i}-D_{i}^{gt}{\|}^{2}}.$$

A calibration template is placed in the scene and used as part of the environment for 3D reconstruction, as shown in Figure 2. The actual dimensions of the calibration template can be used as a priori knowledge as a reconstruction constraint; thus, the scale factor can be recovered. Because in intracavitary surgery, a priori knowledge can be provided by introducing known-size micro instruments or other surgical tools as calibrators, with the actual size and the estimated size of the reconstruction, the scale factor can be calculated; therefore, the size of the reconstructed model at the actual scale can be recovered.

### 2.2. Results and Discussions

To evaluate the 3D reconstruction of endoscopic-vision SLAM, using real data from human body cavities in the research stage has certain limitations and makes it difficult to conduct sufficient testing. Therefore, based on the actual needs and constraints of endoscopic scenarios, equipment, and data acquisition, this paper buildt a simulated endoscopic experimental platform, using a synthetic model to test in a realistic indoor environment, and performed 3D reconstruction of the simulated cavity using an endoscopic monocular SLAM system. In order to recover the scale factor of the endoscopic monocular SLAM system and evaluate its accuracy, this paper combined two feasible methods: co-reconstruction with an intracavitary calibration object and pre-calibration outside the body, based on the minimally invasive surgery scenario. Then, mainly from the perspective of absolute size error, we calculated the measurement error of endoscopic SLAM and use the root-mean-square error (RMSE) to quantify this error value. Moreover, considering that during surgery inspection using laparoscopy, more attention was paid to local measurement information of the target areas or specific lesions; we selected typical parts with prominent features on target organs or soft tissue surfaces in the cavity as lesion areas after obtaining the reconstructed point cloud model. We used a local alignment evaluation method to quantitatively evaluate the 3D reconstruction accuracy of the system from the direction of feature structure size alignment.

This section verifies the effectiveness and reliability of the endoscopic monocular SLAM system in a simulated abdominal puncture exploration by building a simulated human body cavity experimental platform. It mainly focuses on conducting multiple endoscopic monocular SLAM reconstruction experiments around the designed intracavitary accuracy evaluation method and quantitatively evaluating the accuracy of the reconstruction data as well as analyzing and discussing the corresponding results. Thus, it studied the validity of the proposed evaluation method for this scenario and determined the precision of the system. In this section, the relevant experimental hardware configuration of the endoscopic monocular vision SLAM-based 3D reconstruction platform is shown in Table 1.

In the endoscopic 3D reconstruction experiment based on monocular SLAM, this paper simulated a realistic minimally invasive surgery scenario by using a synthetic abdominal cavity box and a straight rod laparoscope on a mechanical experimental platform, as shown in Figure 3a. The abdomen of the simulation box was made of silicone material, and a stomach organ model was placed in the box as the target subject of the intracavitary experiment. A puncture tool could be used to insert the laparoscope from the silicone abdomen for exploration. On the endoscope, we used a 30-degree straight rod laparoscope with a light source, which had a diameter of 10 mm, a front-end monocular camera resolution of 640 × 480, a frame rate of 30fps, and eight LED ring light sources around the lens. Moreover, its 30-degree oblique angle was consistent with the clinical surgery perspective, as shown in Figure 3b. At the same time, in order to restore the dark environment inside the cavity under real laparoscopic surgery as much as possible, we always ensured that there were no other light sources in the experimental environment. We only used the built-in light source of the laparoscope as an illumination condition to simulate shooting inside the abdominal cavity in an almost completely dark laboratory.

This paper uses Zhang’s calibration method to calibrate the intrinsic parameters of the laparoscope used in the experiment. The calibration work uses a 9 × 7 chessboard, with a single black and white square size of 24 mm. As shown in Figure 4a, 25 images were taken from multiple rotation angles through the laparoscope as the calibration image source. Figure 4b shows the detection result of the calibration image, where the green circles are the chessboard corner points extracted in the calibration, and the red crosses are re-projection points, representing the spatial three-dimensional points of the corner points re-projected to the calibration image according to the calibrated intrinsic parameters. They are used to evaluate the accuracy of the calibration results. The smaller the re-projection error, the better the calibration accuracy.

To enhance the calibration accuracy, we used the average pixel error of the images to filter and eliminate individual chessboard pictures, and finally obtain the remaining 15 pictures to calibrate again. After, the calibration was completed; the overall error of its results is shown in Figure 5. The maximum calibration error was 0.26 pixels, and the average calibration error was 0.20 pixel, which met the calibration error requirements. This indicates that the laparoscope has a high calibration precision, and its calibration results have a high credibility.

The corresponding internal reference data of the laparoscope can be obtained through laparoscopic calibration work, and the specific values are shown in Table 2.

Among them, $k_{1}$, $k_{2}$ and $k_{3}$ are radial distortion parameters, and $p_{1}$ and $p_{2}$ are tangential distortion parameters, which are used to correct different degrees of distortion caused by material and production process factors during imaging. In visual SLAM, in order to obtain accurate results, the camera intrinsic parameters are usually treated as known parameters for camera file configuration, and the corresponding parameter files need to be imported before the system runs.

Firstly, for the method of converting coordinate relations with the help of a known calibration template given in Section 2.1, we used a chessboard for auxiliary measurements and placed a chessboard plane target near the target organ tissue in the reconstructed cavity. Since the length of each small square on the chessboard surface was precise, and the laparoscope intrinsic parameters were known, we could obtain the coordinate information of each chessboard corner point according to the projection matrix and transformation relationship between coordinates. Therefore, we could recover the scale factor of the translation vector in the monocular SLAM imaging process and obtain a reconstructed point cloud model with real scale.

In the intracavitary reconstruction error evaluation experiment, our overall shooting condition was in a realistic indoor environment. We turned off all external light sources, and put the “stomach” organ model shown in Figure 6a as the target object into the human abdominal cavity simulation box. By holding the laparoscope, we simulated exploration with its own light source and took image sequence data from multiple angles and positions around the stomach, which served as the original input data for the monocular intracavitary SLAM system. As shown in Figure 6b, a checkerboard grid planar target was placed in the simulation box as a calibrator to participate in the accuracy test of monocular internal cavity SLAM. And in order to further evaluate the accuracy of 3D reconstruction quantitatively, a number of linear segments with more obvious and easy-to-measure characteristic corner points were manually selected for measurement in the internal scene of the simulated box taken by laparoscope, as shown in Figure 6b (red lines numbered as 1, 2, 3 and blue lines numbered as 4, 5, 6). The measurement results of the system reconstruction model were compared with the actual distances obtained from real measurements and the errors were calculated. The selected sample line segments are the numbered sections in the figure.

Figure 7 shows the reconstructed point cloud after obtaining the real scale. Because the surface of the organ model used does not have too many blood vessels or undulating structures compared to the soft tissue of the real internal organs, the blank area of the model surface is too smooth and lacks texture. So, fewer feature points can be extracted, which makes the reconstructed model have corresponding deficiencies and holes. On the other hand, because the laparoscope focuses on the location of the selected target measurement area during the reconstruction process, the acquisition for the right side of the model is not sufficient. However, it can be seen that the point cloud model obtained from the 3D reconstruction is basically able to provide a more correct and detailed representation of the model of the target region and the tessellation target, and the parameters of the selected sample line segments are calculated and measured by this reconstructed model.

The accuracy is calculated from the model by selecting the average measured distance from the source to the target points, and the experimental measurement comparison results are shown in Figure 8. The percentages on the bar graph represent the relative error percentage between the reconstructed data and the real measurement data. The RMSE of the selected local structure dimensions is 2.12 mm, and the root-mean-square value of the relative error is 8.04%.

## 3. In Vitro Pre-Calibration Method for SLAM Reconstruction

The method of placing a calibration template in the scene must take into account the size or volume impact of the calibration template. Therefore, in special cases, some intracavitary spaces are too small and complex, and due to surgical operation or patient’s body reasons, it is impossible to introduce external tools or calibration objects into the cavity. In this case, the method of co-reconstruction using intracavitary calibration objects cannot be used. Based on this problem, this paper designed a method of pre-calibration outside the body, considering recovering the scale information of the monocular SLAM system in the cavity, by pre-calibrating through the laparoscope inserted outside the abdominal cavity. A calibration box for pre-calibration outside the body was made. During surgery, the laparoscope slowly passed through the calibration box before entering cavity. The geometric diagram is shown in Figure 9a. The overall shape of the calibration box is rectangular with circular holes on the front and back sides for the laparoscope to pass through. Feature markers were arranged on four sides inside the box. Figure 9b shows the structure of scene inside the calibration box obtained using a laparoscope.

The inner walls of the calibration box have two parallel planes on top and bottom. Therefore, based on the constraint condition of plane parallelism, by considering the pixel coordinate range and normal relationship of reconstructed map points, map points in the top and bottom plane regions were selected to obtain geometric expressions of two fitted planes. The geometric equations of the top and bottom planes were set as follows:(3)$$\left\{\begin{array}{l} Lx+Jy+Kz+d_{1}=0\\ Lx+Jy+Kz+d_{2}=0 \end{array}\right.$$
where the plane equation parameters L, J, *K*, $d_{1}$, $d_{2}$ are obtained by optimizing the objective function constructed by Equation (4).
(4)$$a=1\underset{L,J,K,d_{1},d_{2}}{\arg\min}\left(\sum_{i=1}^{V}\frac{{\left(Lx_{i}+Jy_{i}+Kz_{i}+d_{1}\right)}^{2}}{L^{2}+J^{2}+K^{2}}+\sum_{j=1}^{W}\frac{{\left(Lx_{j}+Jy_{j}+Kz_{j}+d_{2}\right)}^{2}}{L^{2}+J^{2}+K^{2}}\right)$$

Point selection was repeated for fitting in order to obtain the best parameter estimation results after optimization iteration. Then, the reconstructed estimation value of the distances between the top and bottom planes was further calculated, and traditional measurement methods were used to measure the real distance inside the calibration box. Thus, the scale factor *s* was obtained to recover the real scale of the monocular SLAM-based 3D reconstruction in the cavity.

The experimental scenario for quantitative evaluation using the in vitro pre-calibration method is shown in Figure 10a. Similarly, to be as close as possible to the real laparoscopic operation, it was performed in a dark environment. A calibration box was placed in front of the abdominal puncture site during the experiment, and the laparoscope was slowly pushed into the box to fully acquire information about its internal wall images before entering the internal cavity to obtain image data of the organ and the internal cavity environment.

We chose a particular site on the simulator organ’s surface as the target lesion and measured its dimensions as an evaluation metric. Figure 10b shows the red dots indicating the lesion endpoint that we manually picked in the key frame. We computed the Euclidean distance between the target points in the reconstructed model and compared it with the actual size of 36.04 mm. 

Figure 11 displays the point cloud model after scale restoration. To visualize its results and view the target surface from an optimal angle, we reversed the z-axis of the reconstructed outcome. The model reconstruction accurately captured the features of the organ surface. 

We selected five map points at both ends of the target lesion and performed five trials. Table 3 listed the three-dimensional coordinates of these endpoints. The reconstruction measurement had an RMSE of 2.99 mm and an RMSE rate of 8.30%.

In the real cavity environment, some organs or soft tissues often have uneven, wrinkled and irregular surfaces, which are more complex in spatial shape. Therefore, in the simulation of the cavity, the organ model shown in Figure 12 was used to simulate the soft-tissue surface with more wrinkles, and verify the performance and applicability of the monocular SLAM-based system in the cavity for 3D reconstruction of the cavity target under this situation, and the above two evaluation methods were used to test the system reconstruction measurement accuracy. The blue points in the figure are the assumed endpoints of the target lesion to be measured. The actual measurement value between the two endpoints on the model was 20.80 mm, which was regarded as the true size of the lesion.

The 3D reconstruction effect of soft tissue after restoring the correct scale size by using the two methods of intracavity calibrators and in vitro pre-calibration is shown in Figure 13.

It can be seen that the overall shape is consistent with the real model, and the contours and concave-convex details on the model are also clear, which can allow us to obtain the soft tissue information in the cavity well. The feature points of the local lesion were selected to evaluate the matching degree of the feature structure size, and to calculate the error between the reconstruction measurement results and the actual values.

The measurement data and statistical error analysis results of the 3D reconstruction of the cavity are shown in Table 4. Similarly, five experiments were conducted on the soft-tissue model. According to the data in the table, the RMSE of the local size matching of the target lesion measured by using in-cavity calibration and joint reconstruction was 1.94 mm, and the relative error rate was 9.33%, while the RMSE and error rate of the measurement by using the in vitro pre-calibration method were 2.13 mm and 10.26%, respectively.

By using two different models with different appearance morphologies and features to simulate the target organs and soft tissues in the cavity environment, the monocular SLAM in the cavity successfully realized the 3D reconstruction of the targets in both situations. For the quantitative evaluation of the system reconstruction results, two methods of in-cavity calibration and ex vivo pre-calibration were used to evaluate the reconstruction accuracy from the perspective of feature structure size matching. According to the error statistical data of the two experimental results of simulated organs and simulated soft tissues, it can be seen that the measurement error rate of both methods was around 10%, but the reconstruction model accuracy after restoring the real scale by using the in-cavity calibration template to provide prior knowledge was higher. Therefore, for judging the accuracy of the reconstruction results of the cavity SLAM system in a minimally invasive surgery scenario, introducing feasible calibration objects in the cavity first, such as small flexible rulers, surgical tools and other objects with obvious features and known size to be placed around the target, should be considered for data collection and reconstruction together. But when limited by objective factors such as surgical conditions, and relevant calibration objects cannot be placed in the cavity environment, then the method of in vitro pre-calibration should be chosen, so that the laparoscope can obtain the scale factor during the process of inserting into the cavity.

## 4. Conclusions

In view of the degree of ambiguity of monocular SLAM and the difficulty in obtaining the actual true value of the lumen inspection data, which makes it impossible to quantitatively evaluate the accuracy of the iterative reconstruction of the lumen SLAM through common methods, this paper, starting from the direction of local coincidence, and based on the idea of GCP, designed two quantitative accuracy verification schemes for the reconstruction of intracavity calibrators and in vitro pre calibration. In addition, a simulated intracavity experimental platform was set up for actual measurement. The measurement results of the intracavity calibration reconstruction method in the two schemes were good, and the root-mean-square deviation of the reconstruction measurement in simulated organs and simulated soft tissues was 2.13 mm and 1.94 mm, respectively. Therefore, the feasibility of lumen SLAM and accuracy quantitative evaluation scheme was verified through the actual test of the simulated lumen, and the evaluation results can be used as a reasonable basis to measure the quality of iterative reconstruction of lumen SLAM.

## Figures and Tables

**Figure 1 sensors-23-07188-f001:**
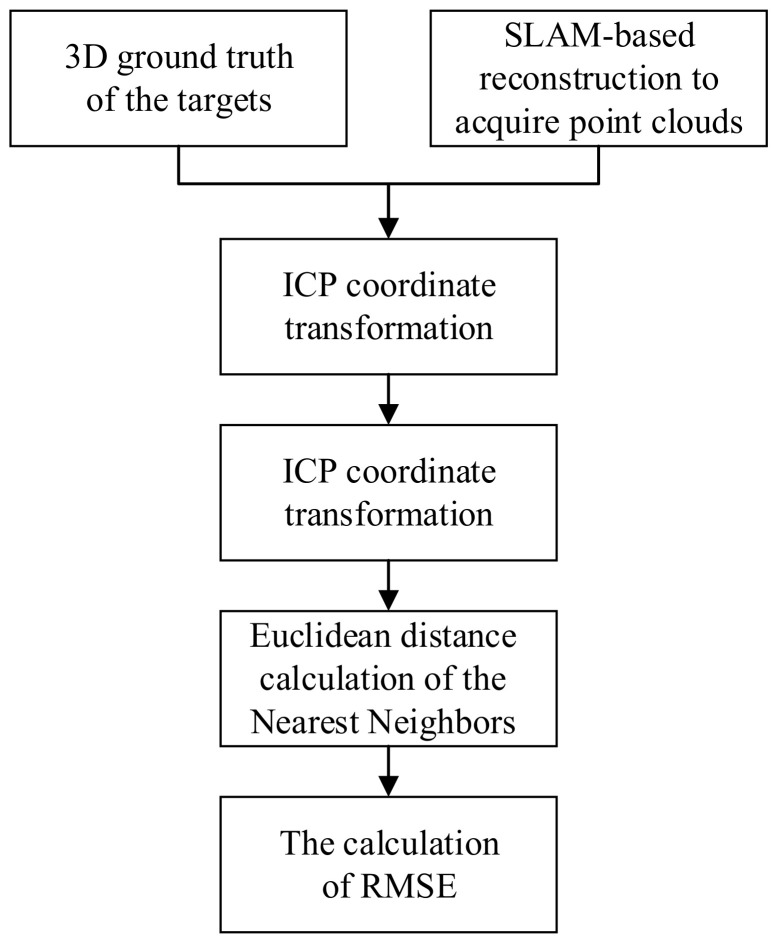
The flowchart of absolute accuracy evaluation for the reconstructed point cloud.

**Figure 2 sensors-23-07188-f002:**
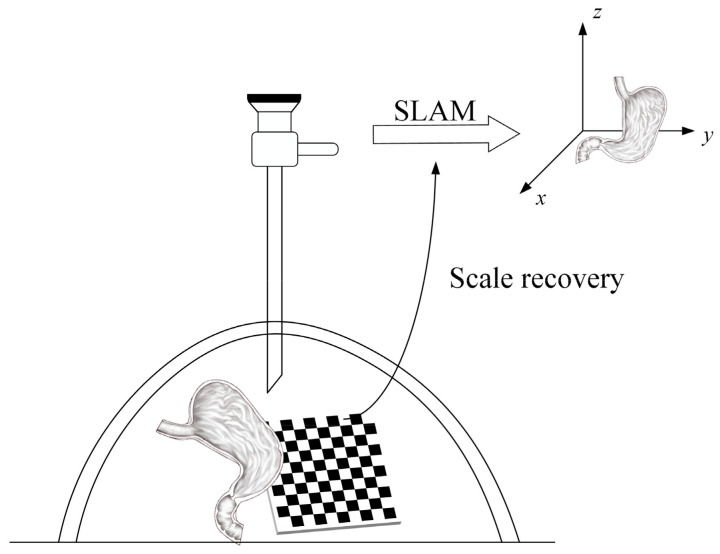
Schematic diagram of the intracavity calibrator.

**Figure 3 sensors-23-07188-f003:**
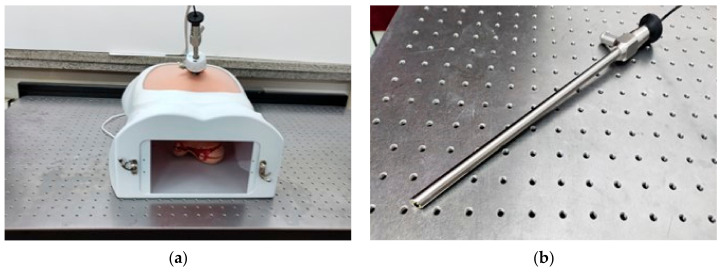
The hardware equipment (**a**) Simulation of internal cavity experimental scenes; (**b**) straight rod laparoscopy.

**Figure 4 sensors-23-07188-f004:**
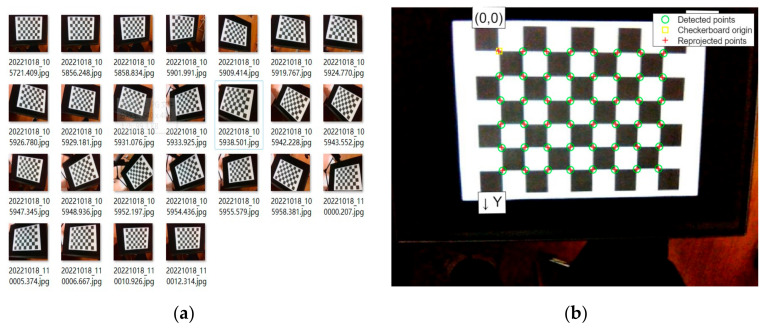
Laparoscopic calibration work: (**a**) Calibrated image sources; (**b**) calibrated test results.

**Figure 5 sensors-23-07188-f005:**
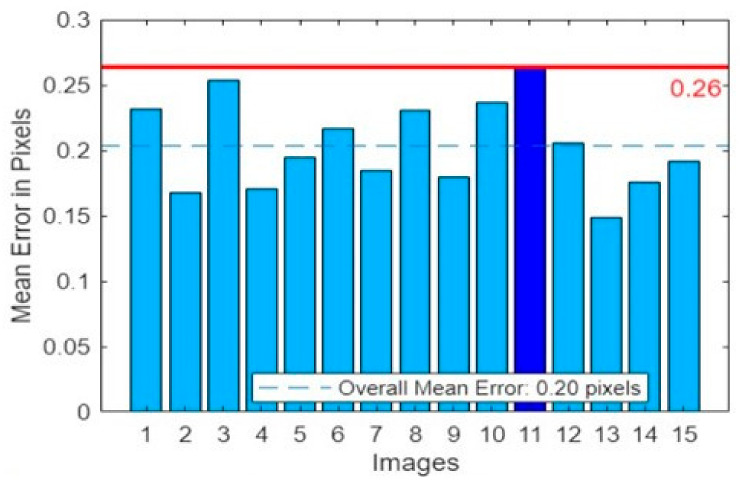
The error in calibration results.

**Figure 6 sensors-23-07188-f006:**
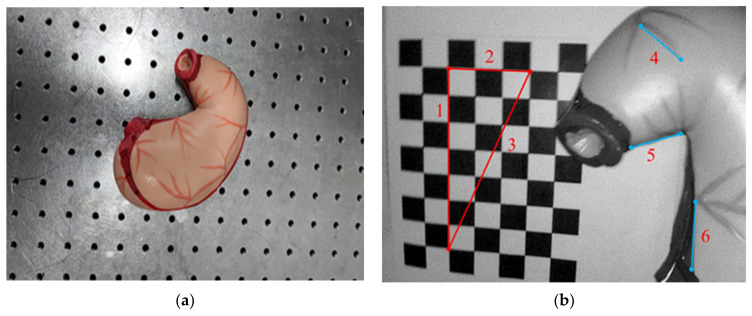
Simulation of experimental objects: (**a**) Photo of organ model; (**b**) test images of the simulated inner cavity.

**Figure 7 sensors-23-07188-f007:**
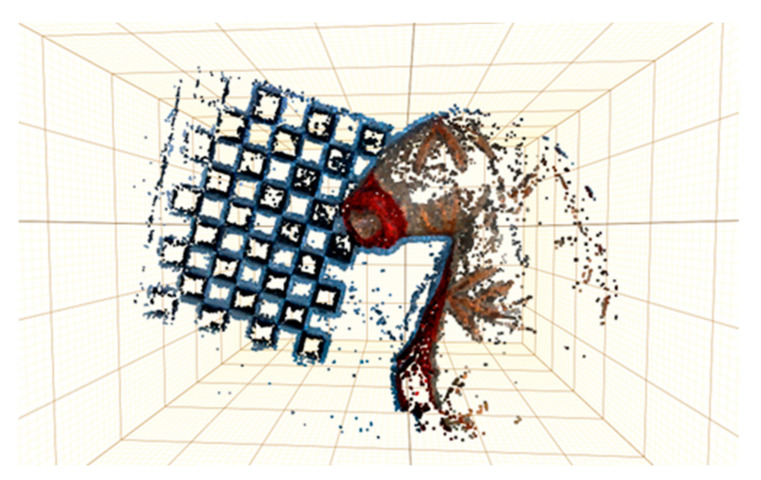
Calibrators and organ reconstruction results.

**Figure 8 sensors-23-07188-f008:**
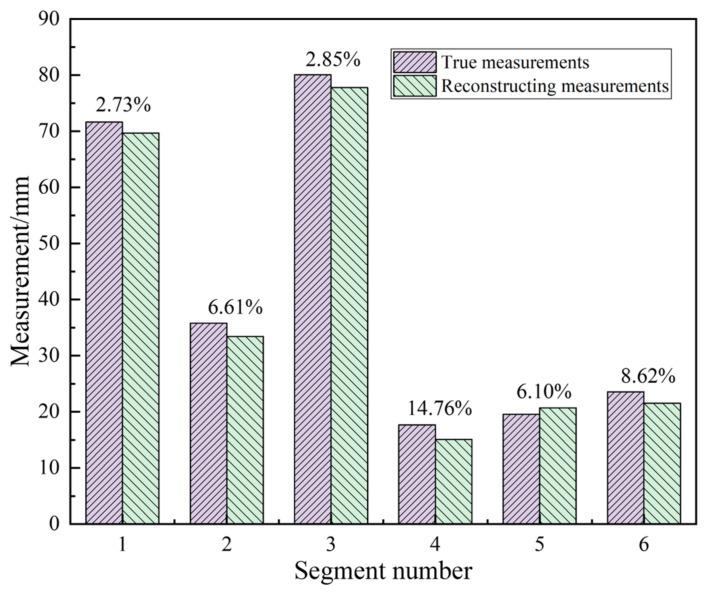
The measurement results of sample line segments.

**Figure 9 sensors-23-07188-f009:**
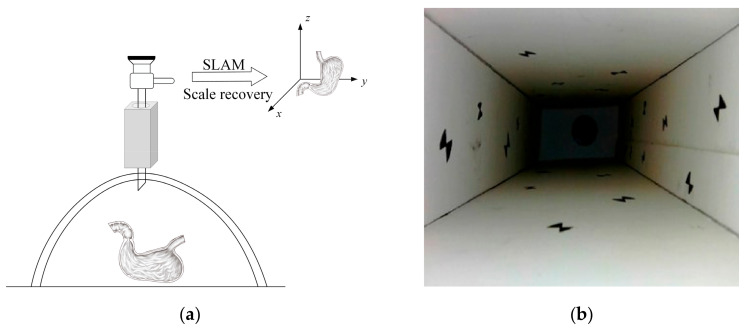
In vitro pre-calibration method: (**a**) Geometric diagram; (**b**) scene inside the calibration box.

**Figure 10 sensors-23-07188-f010:**
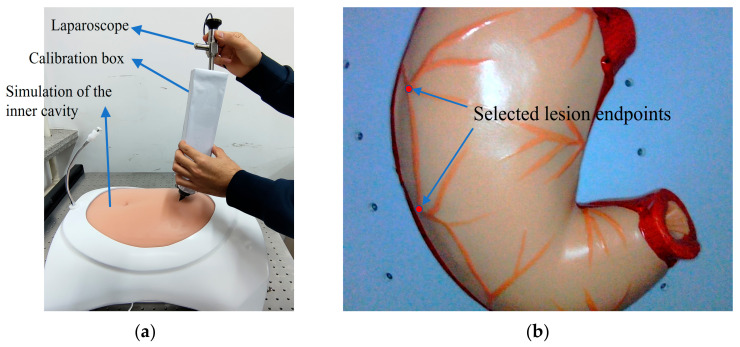
Simulated scene of the in vitro pre-calibration experiment: (**a**) In vitro pre-calibration experimental scenarios; (**b**) the simulated lesion location.

**Figure 11 sensors-23-07188-f011:**
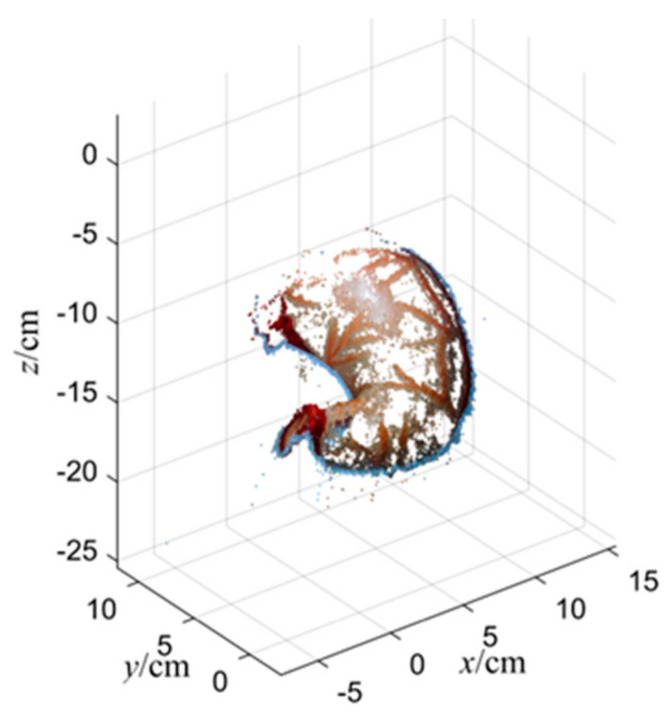
The results of organ reconstruction.

**Figure 12 sensors-23-07188-f012:**
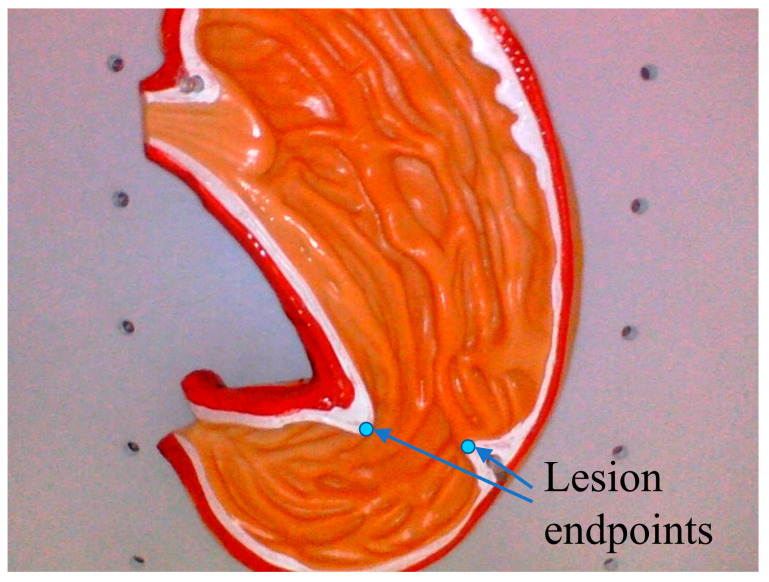
The schematic of simulated soft tissue surface.

**Figure 13 sensors-23-07188-f013:**
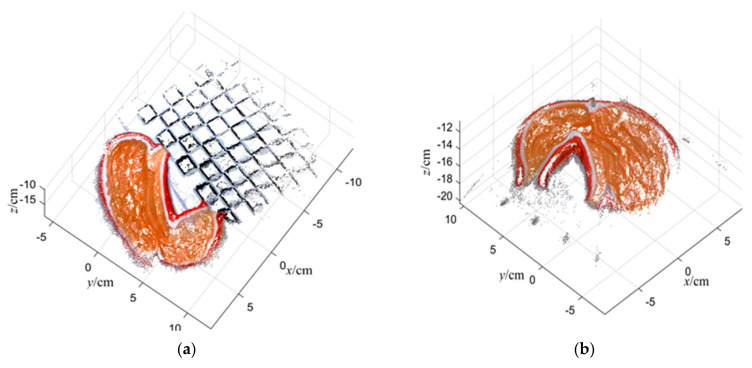
Soft-tissue reconstruction results. (**a**) Results of intracavity calibrator reconstruction method; (**b**) results of the in vitro pre-calibration method.

**Table 1 sensors-23-07188-t001:** The configuration of experimental platform.

Hardware	Quantity
Computers	1
Abdominal Simulation Box	1
Straight rod laparoscopy	1
Mechanical platform	1

**Table 2 sensors-23-07188-t002:** Calibrated data of laparoscopic internal parameters.

$f_{x}$	$f_{y}$	$c_{x}$	$c_{y}$	$k_{1}$	$k_{2}$	$k_{3}$	$p_{1}$	$p_{2}$
620.0051	619.9470	328.6956	241.1636	−0.0252	0.3432	−1.5024	0	0

**Table 3 sensors-23-07188-t003:** The 3D coordinates of simulated lesion endpoints.

Experiment Number	Map Point Coordinates/cm	Euclidean Distance/mm
1	(−5.918, 1.464, 16.131)	39.98
	(−7.542, −0.976, 13.412)	
2	(−5.796, 1.320, 16.078)	39.22
	(−7.495, −0.989, 13.401)	
3	(−5.925, 1.480, 16.156)	38.66
	(−7.431, −0.882, 13.491)	
4	(−5.979, 1.507, 16.173)	38.69
	(−7.347, −0.930, 13.495)	
5	(−5.907, 1.496, 16.104)	38.33
	(−7.321, −0.919, 13.485)	

**Table 4 sensors-23-07188-t004:** The measurement error in soft tissue reconstruction.

Experiment Number	Intracavity Calibrator Reconstruction			In Vitro Pre-Calibration Reconstruction		
	Reconstruction of Measured Values/mm	Absolute Error/mm	Relative Error/%	Reconstruction of Measured Values/mm	Absolute Error/mm	Relative Error/%
1	23.01	2.21	10.63	22.99	2.19	10.53
2	22.25	1.45	6.97	23.04	2.24	10.77
3	22.86	2.06	9.90	23.09	2.29	11.01
4	22.59	1.79	8.61	22.87	2.07	9.95
5	22.90	2.10	10.10	22.65	1.85	8.89
RMSE	/	1.94	9.33	/	2.13	10.26

## Data Availability

Data are unavailable.

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
