# Peer review of "A Novel Evaluation Method for SLAM-Based 3D Reconstruction of Lumen Panoramas"

_sensors, 2023, doi:10.3390/s23167188_

Round 1

Reviewer 1 Report

This is an interesting study. Its approach is sound with the extensive background/introduction. The results are clear.  But the methodology and the results are mixed together,  which negatively impact the clarity of this study.  The conclusion can be improved with a clear goal and clearly described approach/methodology. 

It is clear the entire study included two parts: theoretical development of the evaluation method, and several experimental studies to simulate the actual laparoscopy procedure which intends to test the newly developed evaluation method. All description of  how the experimental studies were designed and conducted should be included in the method section, NOT the results section.  It is convenient for authors to put an experiment and its results together, in particular, when there are several experimental studies to report within a word count limit.  But it is confusing for readers to follow, not mention to understand, what this study is in details.  I strongly suggest authors modify the structure. Specifically, 3 Results from line 275-L446, can be splited into two parts: descriptions of each experimental study in method, while their results remain in Results. 

In addition, a list of abbreviations should be included before the introduction.

Also, The term of accuracy and precision are important in this study. Please follow ISO 5725-1 1994 definition and use these terms consistently.

Some specific comments, questions and suggestions are  list here for your considerations:

L 11: Field of vision? or Field of View?

L13:  in 3D?

L 7: How about "existed" instead of " prior" knowledge?

L21: How about " suggests" instead of " indicates"?

 L65: please clarify what is "monocular SLAM"? Compare to SLAM?

L80: Please clarify "precise multi-sensor calibration"

L82: please clarify " real data label"

L92: What is GCP? please provide full term when it appears in the first time.

L128-131: Something is missing here logically. Please reconsider.

L137: ICP?

L151-153: this definition of congruence is confusing.

L194: abbreviation,  RMSE, is suffice here.

L208: what is " the estimation relationship"?

L228: What is PCL? and what is the limitations of PCL?  What is " PCL point cloud processing"?

L452-456: Typo. Delete.

L487-496: These sentences described this study. They are not the conclusion of this study.

Reviewer 2 Report

The objective of the paper is not clear. I am not sure the authors want to demo the feasibility of 3D reconstruction from laparoscopic images obtained in simulated lumen environment or they try to develop a new quantitative evaluation method of SLAM 3D reconstruction. But in personal opinion, the research approach need to be improved in both case.

First, it is not possible to put a calibration reference such as chessboard used in this paper inside human body. A very compact calibration target may greatly reduce performance of calibration and also induce the risk to be lost inside lumen. Anyway, there is no data to support such idea in current version of the paper.   

Second, in most cases, the visibility of laparoscope is very limited.  It is very rare to able see the whole organ, which is usually shield by neighbour organs or some other structures. The images obtained usually are only cover portion of the organ and not have clear feature as the organ model used in the paper.  While as shown in Fig 9, even for the simulated organ model, only the high contrast part, the deep coloured part near the entrance port of the model, can be reconstructed. I doubt its reference value for real surgical operation. The approach maybe has some impact to mitigate the poor illumination condition in laparoscopy imaging as mentioned by the author. However, besides few sentences description, there is no detailed data or study, under different illumination condition, in the manuscript. The role of the calibration box is not clear. It plays the same role or just alternative format of the calibration reference like the chessboard? If so, why still need chessboard and why it needs to be done just before the probe enter body?  

The evaluation approach used in the paper, eqn 3 and the second eqn 1, are common and not show any new feature in the literature discussed in introduction part.

Obviously, the paper is not seriously prepared. It is full of typo, grammar errors and editing mistakes. E.g.

There is two equation (1) at line 166 and line 275. The introduction part discuss a lot about the advantage and disadvantages of SLAM (bef line 85), while not about the quantitative evaluation challenge of 3D reconstruction. There is also not clear boundary between literature review and methodology used in the paper. Maybe the section 2.1 and 2.2 is the proper introduction to shown the background of the paper?

In line 115 to 116 The author category 3D data capture into manual approach , which is defined as method using tape, callipers, and instrument-based, which is defined as laser scanner, CT scanner etc. I don’t think it is proper way and not able to catch how it is relevant to author’s work?

They forget to delete the template guide in line 453-456 “Authors should discuss the results and how they can be interpreted from the perspective of previous studies and of the working hypotheses. The findings and their implications should be discussed in the broadest context possible. Future research directions may also be highlighted.”

Reviewer 3 Report

The paper "A Novel Evaluation Method on SLAM-based 3D Reconstruction of Lumen Panorama" presents a novel approach that utilizes Simultaneous Localization and Mapping (SLAM) technology for reconstructing laparoscopic scenes. The authors' key innovation is the incorporation of prior knowledge from calibration objects to supplement metric information and resolve scale ambiguity issues, achieved through a calibration step. The paper also quantifies the accuracy of endoscopic reconstruction using local alignment metrics. The results and experiment setup are useful and effectively support the authors' claims, demonstrating improved performance compared to existing techniques.

However, I have some concerns regarding the robustness of the model to noise and small movements of the patient organs. During real experiments, patient organs are not stationary and can move due to factors such as breathing or other disturbances, even if the patient is under anesthesia. The paper does not provide evidence that their model is robust to such movements. Moreover, since a nearby neighbor was used to calculate the point distance between ground truth and reconstructed results, it is unclear how the model would perform if small movements were introduced during testing. For instance, if the air were to be ingested into the Lumen, it could create vibrations in the objects being reconstructed, and this could affect the accuracy of the model. Therefore, I think it would be beneficial for the authors to further investigate the robustness of their model to different types of noise and disturbances, and to evaluate its performance under realistic conditions.

Some small errors,

(1) Acronyms should be mentioned in full words before being used directly. For example, GCP in line 92, ICP in line 137, and PCL in line 228. It would be helpful for readers if the full words or references are provided to aid their understanding. While I could determine that GCP stands for Ground Control Point, I could not find the meanings of the other two acronyms. Therefore, it is important for the authors to provide explanations or references for these acronyms to ensure clarity for readers.

(2)On page 7, lines 274 to 275, the equation appears to be incorrect. Is the first 1 a typo? Additionally, the numbering of equations should be equation 6 but labeled as equation 1. Please ensure that the numbering of equations is accurate for clarity and consistency in the manuscript.

(3) In Page 15, the discussion section, the first paragraph, lines 453 to 455, states, "Authors should discuss the results and how they can be interpreted from the perspective of previous studies and of the working hypotheses. The findings and their implications should be discussed in the broadest context possible. Future research directions may also be highlighted." This appears to be an outline draft or a note to the authors, and it should be removed from the final manuscript to ensure that only the relevant content is included in the discussion section.

Round 2

Reviewer 1 Report

This is a well conceived study with newly proposed method and
experimental studies to validate them. The method proposed
more likely does not work, as evidenced by the design flaws (lack of control,
small sample size, 2D chess board for 3D problem...) of the experimental studies.
We need better approaches and better designed studies.
Even so, this paper has its merits to be published so as to allow others researchers
to continue studies  in this direction.

Thanks Authors' responses to my previous comments and questions.
The revised version addressed some of my concerns.

One major issue remains:

The objective/aim ( L 140-149) of this study is not clear,
and has certain conflicts with Conclusions (L 480-494).
They Need further clarification/modification.

Also, all sentences in above two paragraphs are in present tense.
Wondering why so?

The story-line of this paper is clear: either one ( as in title)
or two ( as in the text) methods proposed,
several experimental studies were conducted, and proved
the proposed methods are accurate within a reasonable range.

Therefore, the objective/aim of this study is:
(1) to propose new method;
(2) to design and conduct the experimental studies;
(3) to prove the results of the new method is accurate. 

Therefore, the present tense may not apply.
For example,  L147-149: "The experimental results demonstrate that the
reconstructed model restores the true scale information, reveals the overall accuracy
 of
laparoscopic SLAM system, and offers technical guidance for subsequent applications."
It does not make sense in this paper. If the above sentence holds, why conduct this study ?
Simply repeat the same sentence at both objective and
conclusion does not work.

Similarly, the conclusion should like: the new method proposed.
The experimental studies conducted 
successfully.
The results proved the new method is accurate with an accuracy of ....  
Or, it may use the present tense: The new method is accurate in terms of ....

Please rewrite objective/ aim and conclusions with higher clarity
and consistency.

L208. Based upon your response, it is " estimating relationship",
not " estimation relationship". please confirm.

PCL stands for "point cloud processing", or for " Point Cloud Library"?

Overall, please proofreading and improve the consistency.
Please do check the grammar, in particular, the tense of sentences
properly.

Reviewer 2 Report

As mentioned in Arthur’s reply, the 3D reconstruction method has been proposed and evaluated qualitatively in one of author's early publication, this paper is "To further quantify" such method. The major weakness of the paper is that limited data points may not enough to support their conclusions.

The author provides two groups of data. First group data is obtained using chess board sample as shown in Fig.7. The sample size is quite limited. From 25 pictures obtained in experiments and only 15 pictures are selected for evaluation. The rest 10 pictures, 40% samples, are "filtered and eliminated". The generality and reliability of such approach are questionable. At least, a more detailed discussion should be provided to justify such filtering of the data.

Second group of data are obtained from an artificial organ model. Only single lesion from very limited area is selected and the measurements are only repeated for 5 times as shown in table 3 and table 4. Although it has potential to prove the quantification capability of proposed method, I strongly suggest author consider to either increase the sample size or utilize more general and fair lesion selection criteria to improve the impact and fairness of the data.    

Furthermore, section 2 seems not relevant. I don't see any congruence evaluation results in later experimental section, suggest removing it.

Section 3.2  In vitro pre-calibration method. I don't see any necessity of such external calibration must be done at the time instance just before the surgery. Does author hint the calibration results vary with time of execution or the manner of the scope approaching? I will suggest removing the in vitro feature and just use "external pre-calibration" instead. Alternatively, a detailed discussion or explanation on the role and necessity of installing such box directly on the laparoscope is needed.
